# AI-Assistance Body Composition CT at T12 and T4 in Lung Cancer: Diagnosing Sarcopenia, and Its Correlation with Morphofunctional Assessment Techniques

**DOI:** 10.3390/cancers17193255

**Published:** 2025-10-08

**Authors:** Maria Zhao Montero-Benitez, Alba Carmona-Llanos, Rocio Fernández-Jiménez, Alicia Román-Jobacho, Jaime Gómez-Millán, Javier Modamio-Molina, Eva Cabrera-Cesar, Isabel Vegas-Aguilar, Maria del Mar Amaya-Campos, Francisco J. Tinahones, Esther Molina-Montes, Manuel Cayón-Blanco, Jose Manuel García-Almeida

**Affiliations:** 1Departament of Endocrinology and Nutrition, Jerez Universitary Hospital, 11407 Jerez de la Frontera, Spain; mariaz.montero.sspa@juntadeandalucia.es (M.Z.M.-B.); acarmonallanos@correo.ugr.es (A.C.-L.); mancarmac522@gmail.com (M.C.-B.); 2Instituto de Investigación e Innovación Biomédica de Cádiz, Puerta del Mar University Hospital, 11009 Cádiz, Spain; 3Departament of Nutrition and Food Sciencie, University of Granada, 18071 Granada, Spain; memolina@ugr.es; 4Departament of Endocrinology and Nutrition, Virgen de la Victoria University Hospital, 29010 Málaga, Spain; isabel.mva13@gmail.com (I.V.-A.); mariadelmarac2@gmail.com (M.d.M.A.-C.); fjtinahones@uma.es (F.J.T.); josem.garcia.almeida.sspa@juntadeandalucia.es (J.M.G.-A.); 5Instituto de Investigación Biomédica y Plataforma en Nanomedicina-IBIMA Plataforma BIONAND, 29590 Málaga, Spain; jaimegomezmillan@gmail.com; 6Department of Endocrinology and Nutrition, Quironsalud Málaga Hospital, 29004 Málaga, Spain; 7Department of Medicine and Dermatology, Málaga University, 29016 Málaga, Spain; 8Department of Radiación Oncology, Virgen de la Victoria University Hospital, 29010 Málaga, Spain; alicia.roman.sspa@juntadeandalucia.es; 9Departament of Endocrinology and Nutrition, Sureste University Hospital, 28500 Arganda del Rey, Spain; javier.modamio@salud.madrid.org; 10Department of Neumology, Virgen de la Victoria University Hospital, 29010 Málaga, Spain; evab.cabrera.sspa@juntadeandalucia.es; 11Centro de Investigación Biomédica en Red de la Fisiopatología de la Obesidad y la Nutrición (CIBEROBN), Carlos III Health Institute (ISCIII), University of Málaga, 29010 Malaga, Spain; 12Institute of Nutrition and Food Technology (INYTA) ‘José Mataix’, Biomedical Research Centre, University of Granada, 18071 Granada, Spain; 13Instituto de Investigación Biosanitaria ibs.GRANADA, 18071 Granada, Spain; 14CIBER de Epidemiología y Salud Pública (CIBERESP), 28029 Madrid, Spain

**Keywords:** sarcopenia, lung cancer, computed tomography, skeletal muscle index, bioelectrical impedance analysis, morphofunctional assessment

## Abstract

**Simple Summary:**

Muscle loss, known as sarcopenia, is a common problem in people with lung cancer and can negatively affect their recovery and overall health. However, diagnosing it early in routine hospital visits is difficult. This study looked at whether existing chest scans (Computed Tomography (CT) scans) performed for cancer diagnosis could also be used to evaluate muscle loss. We analyzed two parts of the spine, namely T12 and T4, in 80 lung cancer patients and compared them with other tools available for the assessment of nutrition and muscle function. We found that images from the T12 level were especially helpful in spotting patients with low muscle mass or sarcopenia. By combining this with a simple physical test, we improved the accuracy further. These findings suggest that doctors could use routine CT scans to detect muscle problems early, helping to guide better care and nutrition planning without needing further tests.

**Abstract:**

Background: Sarcopenia and low muscle mass are prevalent and prognostically relevant in patients with lung cancer, yet their diagnosis remains challenging in routine clinical practice. Opportunistic assessment using computed tomography (CT) has emerged as a valuable tool for body composition evaluation. We aimed to assess the utility of thoracic CT at T12 and T4 levels in identifying sarcopenia and low muscle mass and explore their correlation with morphofunctional tools such as bioelectrical impedance vector analysis (BIVA), nutritional ultrasound (NU), and functional performance tests. Methods: In this prospective observational study, 80 patients with lung cancer were evaluated at diagnosis. Body composition was assessed using BIVA-, NU-, and CT-derived parameters at T12 and T4 levels. Functional status was measured using the Timed Up and Go (TUG) and 30-Second Chair Stand Test. Sarcopenia was defined according to EWGSOP2 criteria. Results: Sarcopenia was identified in 20% of patients. CT-derived indices at T12CT demonstrated better diagnostic performance than T4CT. For detecting low muscle mass, the optimal SMI cut-off values were SMI_T12CT < 31.98 cm^2^/m^2^ and SMI_T4CT < 59.05 cm^2^/m^2^ in men and SMI_T12CT < 28.23 cm^2^/m^2^ and SMI_T4CT < 41.69 cm^2^/m^2^ in women. For sarcopenia diagnosis, the values were SMI_T12CT < 24.78 cm^2^/m^2^ and SMI_T4CT < 57.23 cm^2^/m^2^ in men and SMI_T12CT < 21.24 cm^2^/m^2^ and SMI_T4CT < 49.35 cm^2^/m^2^ in women. A combined model including SMI_T12CT, RF_CSA, and the 30 s squat test showed high diagnostic accuracy (AUC = 0.826). In multivariable analysis, lower SMA_T12CT was independently associated with risk of sarcopenia (OR = 0.96, 95% CI: 0.92–0.99, *p* = 0.022), as were older age (OR = 1.23, 95% CI: 1.07–1.47, *p* = 0.010) and fewer repetitions in the 30 s squat test (OR = 0.78, 95% CI: 0.63–0.91, *p* = 0.007). Conclusions: CT-derived body composition assessment, particularly at the T12 level, shows good correlation with morphofunctional tools and may offer a reliable and timely alternative for identifying sarcopenia and low muscle mass in patients with lung cancer.

## 1. Introduction

Lung cancer (LC) remains one of the most common cancers globally and the leading cause of cancer-related deaths, accounting for approximately 1.8 million deaths in 2022, with the highest rates in Europe, Asia and North America [1,2,3]. Despite improvements in diagnosis and treatment, prognosis remains poor for many patients. Over the past decade, the clinical relevance of sarcopenia in cancer patients has gained increasing attention. Sarcopenia (Sc) is common in individuals with various primary tumors, particularly in lung cancer, where its estimated prevalence ranges from 40% to 60% [4,5,6]. Numerous studies have identified Sc as an independent risk factor for worse postoperative outcomes, reduced tolerance to therapy, and decreased survival in LC patients [4,5,6,7,8].

To date, there is no international consensus of Sc’s definition. The most widely used definition for this condition has been established by the European Working Group on Sarcopenia in Older People (EWGSOP). According to the latter, Sc is a syndrome characterized by progressive and generalized loss of skeletal muscle mass, muscle strength and performance status [4,5,6,7,8,9]. Physical performance may be evaluated with a timed up and go test (TUG) and the six-minute walk test (6 MWT). With the growing recognition of the importance of muscle assessment, various imaging methods have been introduced to evaluate muscle quantity, including nutritional ultrasound (NU), bioelectrical impedance vector analysis (BIVA) and computed tomography (CT).

CT stands out as the gold standard for muscle mass evaluation, typically performed at the third lumbar vertebra (L3). The cross-sectional area (CSA) at L3 offers reliable estimates of total skeletal muscle mass [5,6,7,8,9,10,11,12,13]; however, most LC staging CT scans do not reach this anatomical landmark.

In response, alternative vertebral levels such as T12 and L1, routinely included in thoracic CT scans, have been investigated for Sc assessment. Multiple studies have demonstrated strong concordance and correlation between T12-derived SMI and standard L3-based measurements, supporting the validity of these alternative thoracic levels for Sc assessment [11,12,13]. In fact, a machine learning-based model using CT measurements at the T12 level demonstrated high predictive accuracy for Sc in LC patients [12]. Machine learning and semi-automated segmentation tools have further enhanced the speed, precision and standardization of muscle quantification from CT imaging in oncology settings [12].

Thus, T12 represents a practical alternative for muscle mass assessment in LC patients when L3-level scans are unavailable. This approach can facilitate the non-invasive identification of Sc without extra imaging procedures, supporting personalized risk stratification and treatment planning [8,10,14]

This study aims to assess the clinical utility of opportunistically acquired thoracic CT images—specifically at the T12 and T4 vertebral levels—in the identification of sarcopenia and low muscle mass in patients with lung cancer. Given that thoracic CT scans are routinely performed as part of LC staging and follow-up, leveraging these existing images for body composition analysis may offer a cost-effective, non-invasive, and time-efficient alternative to traditional methods. Furthermore, this study investigates the correlation between CT-derived muscle indices at T12 and T4 and other morphofunctional assessments, including bioelectrical impedance vector analysis (BIVA), nutritional ultrasound (NU), the timed up and go test (TUG), and the 30 s chair stand test, which are increasingly recognized as reliable tools for evaluating both muscle quantity and functional status. By integrating anatomical imaging with functional and bioimpedance-based assessments, this study aims to contribute to the development of a multidimensional, clinically applicable approach to the early detection of sarcopenia in LC patients.

## 2. Materials and Methods

### 2.1. Setting of the Study

A prospective, observational, single-centre study of routine clinical practice was conducted in the Endocrinology and Nutrition Unit of the Virgen de la Victoria University Hospital. A formal sample size calculation was performed prior to study initiation. Assuming a moderate effect size (Cohen’s d = 0.6), an alpha error of 0.05, and a beta error of 0.2 (power = 80%), a minimum of 72 patients was required to detect relevant differences between sarcopenic and non-sarcopenic individuals. To ensure statistical power and account for potential attrition, a total of 80 participants were enrolled. The study sample comprised patients with lung cancer at various stages, identified through the hospital’s tumor board at the time of diagnosis, who were scheduled to undergo oncological treatment (surgery, chemotherapy, radiotherapy, and/or immunotherapy). Lung cancer subtypes were determined by histological examination and classified as non-small cell lung cancer (NSCLC), small cell lung cancer (SCLC), or other less common histologies, according to standard oncological criteria and pathology reports. This classification is reflected in Table 1.

All participants provided their informed consent for inclusion prior to their engagement in the study. The investigation conformed to the principles outlined in the Declaration of Helsinki and obtained endorsement from the Ethics Committee of Málaga on 25 June 2023 (reference number 1085-N-23).

All individuals enrolled in our investigation satisfied the eligibility criteria, which necessitated a validated lung cancer diagnosis preceding treatment, the provision of signed informed consent for participation, and the execution of a CT scan within one month prior to the initial nutritional evaluation. Furthermore, none of the exclusion criteria were applicable, such as declination of participation, inability to conduct BIVA measurements due to factors including ethnicity-related constraints, extensive dermatological lesions, fluid extravasation, localized hematomas, amputations, or a life expectancy estimated to be less than three months (Appendix A).

Relevant clinical data were collected during the initial visit through the electronic medical record. This assessment included functional status (using the ECOG scale) and parameters related to nutritional status. As part of the morphofunctional evaluation, systematic screening for Sc was performed using the EWGSOP2 criteria, and skeletal muscle mass and its distribution were analyzed, given their association with clinical outcomes in oncology patients.

### 2.2. Anthropometric and Body Composition Assessments

#### 2.2.1. Bioelectrical Impedance Vector Analysis

Body composition (BC) was evaluated utilizing a 50 kHz phase-sensitive impedance analyzer (Nutrilab TM Whole Body Bioimpedance Vector Analyzer, AKERN, Florence, Italy), which delivered a current of 800 μA [15,16] through tetrapolar electrodes strategically positioned on the right hand and foot. Measurements of body weight and height were conducted, and Bioelectrical Impedance Vector Analysis (BIVA) assessments were performed with subjects in a supine position following a five-minute rest period [15,17]. The phase angle (PhA) was computed and subsequently standardized against age and sex-matched reference data derived from a cohort of healthy Italian adults [16,18].

The technical precision of the BIVA apparatus was systematically assessed on a daily basis utilizing a precision track, wherein all recorded values of resistance (R) and reactance (Xc) consistently fell within ±1 Ω of the reference value of 385 Ohm. The In Vivo reproducibility demonstrated a coefficient of variation (CV) ranging from 1% to 2% for both R and Xc [17].

#### 2.2.2. Nutritional Ultrasound^®^

Ultrasonographic evaluation of the rectus femoris quadriceps muscle (RF-CSA) within the lower extremity was conducted in a supine orientation utilizing a 10–12 MHz transducer and a multifrequency linear array (Mindray Z60, Madrid, Spain) administered by qualified professionals. In a supine orientation, we assessed the anteroposterior muscle thickness at the lower third level extending from the superior pole of the patella to the anterosuperior iliac spine. The parameters recorded included the rectus femoris axis (RF-Y-axis and RF-X-axis), circumference (RF-CIR), cross-sectional area (RF-CSA), and subcutaneous fat of the leg (L-SAT) [19]. Each measurement was repeated three times to obtain an average value.

For the assessment of abdominal adipose tissue, we conducted measurements at the midpoint between the xiphoid process and the umbilicus to quantify T-SAT (total subcutaneous abdominal fat), S-SAT (superficial subcutaneous abdominal fat), and VAT (preperitoneal or visceral fat) in centimeters.

#### 2.2.3. Functional Assessment

Handgrip strength (HGS) was evaluated utilizing a JAMAR hand dynamometer (Asimow Engineering Co., Los Angeles, CA, USA) with subjects positioned in a seated posture and the dominant arm’s elbow maintained at an angle of 90 degrees of flexion. Each subject executed three maximal isometric contractions, and the median value was documented [20,21].

The TUG test quantified the duration, expressed in seconds, necessary for a subject to rise from a chair, traverse a distance of three meters, execute a turn, return to the chair, and subsequently assume a seated position [21].

The sit-to-stand (STS) test involves rapid standing up and sitting down times consecutively. In this test, the participants start with their arms crossed over their chest, sitting on a chair without armrests, with their hip and knee joints at 90°. It records the number of times an individual can sit and stand within 30 s [22,23].

#### 2.2.4. Computed Tomography at T12 Level by FocusedON^®^

For T12-CT, axial images obtained at the corresponding vertebral level were derived from computed tomography (CT) scans conducted in a clinical environment within a one-month timeframe prior to the initial nutritional evaluation. At this specific level, the most pertinent musculature encompasses the paravertebral muscles (erector spinae, comprising the iliocostalis, longissimus, and spinalis) along with the quadratus lumborum, as well as the muscles constituting the abdominal wall (external oblique, internal oblique, transversus abdominis, and rectus abdominis). Furthermore, at elevated sections of T12, the dorsal musculature, including the latissimus dorsi, serratus posterior inferior, and intercostal muscles, can be discerned, which are integral to the respiratory process. Although the psoas muscle is more conspicuously delineated at inferior levels, such as L3, its origin may be discerned in certain instances at T12.

The evaluation of skeletal muscle and abdominal adipose tissue areas encompassed the skeletal muscle area (SMA, measured in cm^2^ and %), and the skeletal muscle index (SMI, articulated in cm^2^/m^2^, adjusted for patient height), for both T12 (SMA_T12CT and SMI_T12CT) and T4 (SMA_T4CT and SMI_T4CT). Supplementary variables comprised intramuscular adipose tissue area (IMAT, in cm^2^ and %), subcutaneous fat area (SFA, in cm^2^ and %), and visceral fat area (VFA, in cm^2^ and %). The mean attenuation for each delineated tissue was quantified in Hounsfield units (HU).

CT images focusing on the T12 and T4 vertebral levels were subjected to analysis utilizing FocusedON® software (pilot version, ARTIS Development, Barcelona, Spain), https://focusedon.es (accessed 20 June 2025), which constitutes a sophisticated automatic tissue segmentation tool developed by ARTIS Development and grounded in artificial intelligence methodologies. This software facilitates initial segmentation through artificial intelligence, thereby enabling clinicians to amend and authenticate the segmentation prior to the finalization of the data. The assessment of tissue density was conducted based on its mean HU value employing standardized thresholds: −29 to 150 HU for skeletal muscle, −190 to −30 HU for subcutaneous adipose tissue, and −150 to −50 HU for visceral adipose tissue.

#### 2.2.5. Assessment of Sarcopenia

Sarcopenia (Sc) was diagnosed in accordance with the European Working Group on Sarcopenia in Older People (EWGSOP2) criteria [24,25], which emphasize a staged approach for case finding, diagnosis, and severity assessment. The diagnostic algorithm begins with the identification of probable Sc, defined by reduced muscle strength. In this study, handgrip strength (HGS) was used as the primary indicator of muscle function, with threshold values set at <27 kg for men and <16 kg for women, as established by the EWGSOP2 consensus.

When low HGS was observed, a second step was conducted to confirm Sc by assessing muscle quantity. Muscle mass was measured using BIVA, a validated technique in oncologic settings. A diagnosis of confirmed Sc was made if appendicular skeletal muscle mass (ASMM) was below 20 kg in men or 15 kg in women, or if the appendicular skeletal muscle mass index (ASMI) was below 7.0 kg/m^2^ in men or 5.5 kg/m^2^ in women.

Furthermore, although not required for the base diagnosis, physical performance tests such as the TUG test were used to assess Sc severity. According to EWGSOP2, the coexistence of low muscle strength, low muscle mass, and impaired physical performance defines severe Sc. This multicomponent assessment underscores the relevance of evaluating not only quantitative muscle loss but also its functional implications in clinical outcomes among lung cancer patients.

#### 2.2.6. Statistical Analysis

Statistical analysis was performed utilizing JAMOVI (version 2.3.28 for macOS). Descriptive statistics included continuous variables with a normal distribution, presented as means and standard deviations, and categorical variables, expressed as percentages. The normality of quantitative variables was assessed prior to hypothesis testing.

To compare morphofunctional and clinical parameters between groups (sarcopenic vs. non-sarcopenic), Student’s *t*-test was used for normally distributed variables, while the Mann–Whitney U test was applied for non-normally distributed data. Categorical variables were compared using the chi-square or Fisher’s exact test (<5 observations), as appropriate.

The diagnostic performance of skeletal muscle parameters derived from both T12 and T4 vertebral levels—specifically, SMA_T12CT, SMI_T12CT, SMA_T4CT, and SMI_T4CT—was assessed using receiver operating characteristic (ROC) curve analysis. Optimal cut-off points for identifying low muscle mass and Sc were determined by maximizing the Youden index. The area under the curve (AUC), along with sensitivity and specificity values for each parameter, was calculated to quantify discriminative capacity and guide clinical interpretation.

Correlation analyses among CT-derived indices, BIVA parameters, ultrasound measurements, and functional performance tests were conducted using Pearson’s correlation coefficient for normally distributed variables or Spearman’s rank correlation for non-normally distributed variables. Variables were categorized into muscle and adipose tissue groups for interpretation. The internal consistency of measurement scales was verified using Cronbach’s alpha. A *p*-value < 0.05 was considered statistically significant. Additionally, multivariable logistic regression was applied to evaluate Sc predictors, reporting odds ratios (OR) and 95% CI.

## 3. Results

### 3.1. Demographic, Clinicopathological and Body Composition Characteristics Between Sarcopenic and Non-Sarcopenic According to EWGSOP2 Criteria

This study included 80 patients, of whom 16 (20.0%) were identified as sarcopenic. The overall proportion of male patients was 70.0% (71.9% in the non-sarcopenic group vs. 62.5% in the sarcopenic group) (Table 1). Sarcopenic patients were older (71.8 ± 7.90 vs. 64.8 ± 9.58 years) and had a lower BMI (23.2 ± 2.96 vs. 27.4 ± 4.93 kg/m^2^) than their counterparts. A lower proportion of sarcopenic individuals underwent surgery (5.0% vs. 27.5%), received radiotherapy (5.1% vs. 26.6%), chemotherapy (5.1% vs. 40.5%), or immunotherapy (3.8% vs. 22.8%). Regarding tumor classification, non-small cell lung cancer (NSCLC) was more frequent in the non-sarcopenic group compared to the sarcopenic group (54.4% vs. 12.7%). Although differences did not reach statistical significance, sarcopenic patients tended to present with more advanced tumor stages. Moreover, sarcopenic patients presented worse functional status, with ECOG 3 observed only in this group (2.8% vs. 0%).

Table 2 presents the diagnostic components of sarcopenia following the EWGSOP2 algorithm. Muscle strength, as measured by handgrip strength (HGS), was significantly lower in women compared to men (20.8 ± 9.18 vs. 34.1 ± 10.2 kg; *p* < 0.001). Mean ASMM was significantly reduced in women (15.2 ± 2.9 kg) compared to men (21.9 ± 4.14 kg; *p* < 0.001). Notably, 22.5% of the total sample had low ASMM, while 48.8% exhibited low ASMI values.

When combining low ASMM and/or low ASMI, 53.8% of patients were classified as having low muscle mass. Dynapenia, defined by low HGS, was observed in 23.8% of subjects. The integrated diagnosis of Sc (co-existence of dynapenia and low muscle mass) was identified in 20% of the sample.

Table 3 shows a comparative analysis of morphofunctional and body composition parameters between sarcopenic and non-sarcopenic patients. Individuals with Sc exhibited significantly lower values across multiple domains.

In the BIVA, sarcopenic patients had significantly lower BMI (23.2 ± 2.96 vs. 27.4 ± 4.93; *p* = 0.002) and PA (4.06 ± 0.93 vs. 4.72 ± 0.86; *p* = 0.009) than the non-sarcopenic ones. Additionally, significant decreases were observed in BCM, ASMM, FFM, and TBW values among patients with Sc, indicating a marked deterioration of body composition. A higher NAK ratio was also noted in sarcopenic individuals (1.47 ± 0.40 vs. 1.24 ± 0.26; *p* = 0.006). Regarding NU, significant differences were observed between both groups in RF_CSA (2.65 ± 0.73 vs. 3.83 ± 1.30; *p* = 0.002) and RF_Y_Axis. Lower values were also found for L_SAT, T-SAT, and VAT, reflecting reduced adipose tissue compartments in sarcopenic patients. No statistically significant differences were found in TUG performance between the two groups (*p* = 0.544), but there were differences in the 30 s squat test (*p* < 0.036).

Table 4 presents a comparative analysis of body composition parameters derived from thoracic CT scans at T4 and T12 levels in patients with and without Sc. Among the T4-level metrics, only the skeletal muscle area (SMA_T4CT) showed a statistically significant difference between groups (147.7 ± 39.3 cm^2^ vs. 123.7 ± 31.1 cm^2^; *p* = 0.033). In contrast, the T12 level demonstrated a stronger discriminative capacity. Sarcopenic patients exhibited significantly lower skeletal muscle area and muscle index at the T12 level (SMA_T12CT: 62.09 ± 12.65 vs. 82.64 ± 26.29 cm^2^, *p* = 0.008; SMI_T12CT: 23.03 ± 4.21 vs. 29.12 ± 8.48 cm^2^/m^2^, *p* = 0.015). Furthermore, a reduction in subcutaneous adipose tissue was observed in terms of area (SAT_T12 cm^2^: 71.59 ± 32.32 vs. 108.44 ± 68.13 cm^2^, *p* = 0.047).

### 3.2. Correlation Between Morphofunctional Parameters (Muscle and Fat Tissue) and Sarcopenia Criteria

The correlation heatmap revealed moderate to strong positive associations among morphofunctional variables, particularly between SMI, BCM, RF_CSA, and HGS. All variables showed negative correlations with Sc status, with the strongest being SMI (*r* = −0.68), followed by HGS (*r* = –0.59), and SMI_T4CT (*r* = −0.55) and SMI_T12CT (r = −53), (Figure 1). The internal consistency of the scale was acceptable, with a Cronbach’s α of 0.765, supporting the reliability of these parameters as a composite construct for assessing muscle health.

Following the evaluation of muscle-related metrics, we explored the relationship between adiposity indicators derived from different morphofunctional assessment techniques. The heatmap shows moderate to strong correlations between adiposity measures from BIVA (FMI), ultrasound (L-SAT, T-SAT), and CT (SAT, VAT, IMAT). Strong associations were found between FMI and SAT_T4CT (r = 0.91), and between ultrasound (T-SAT and L-SAT) and SAT_T12CT (r = 0.66–0.72), indicating good concordance across techniques. The internal consistency of the adiposity parameters was acceptable (Cronbach’s α = 0.708), supporting their coherence as fat-related indicators (Figure 2).

### 3.3. Cut-Off Points for Parameters of Low Muscle Mass and Sarcopenia Criteria

Table 5 displays the diagnostic performance of both absolute and height-adjusted skeletal muscle metrics—SMA and SMI—obtained from thoracic CT at the T4 and T12 vertebral levels, stratified by sex. CT-derived muscle parameters showed moderate to high diagnostic accuracy for identifying low muscle mass in lung cancer patients. Cut-off values are provided separately for men and women, allowing for tailored interpretation across both anatomical sites. In particular, SMA_T12CT in women achieved an AUC of 0.791, and SMA_T12CT in men also showed favorable accuracy (AUC = 0.772) (Figure 3). At the T12 level, SMI yielded the highest AUCs in both sexes (0.733 in men, 0.802 in women), reinforcing the value of this metric for identifying low muscle mass. Overall, these findings highlight the utility of both SMA and SMI, with the T12 level.

Table 6 summarizes the predictive performance of SMA) and SMI at T4 and T12 levels for the diagnosis of Sc based on EWGSOP2 criteria. As with low muscle mass, sex-specific cut-off values are provided for both anatomical sites and muscle metrics. While overall AUC values were modest, SMI_T12CT in men showed the most balanced diagnostic profile (AUC = 0.653), with high sensitivity (82.5%) and positive predictive value (89.19%). In women, both SMA_T12CT and SMI_T12CT achieved high sensitivity (83.33%), the overall AUC values remained modest. These AUC results indicate that the discriminative ability of these CT-derived parameters to distinguish sarcopenic from non-sarcopenic patients is limited, and their clinical applicability for accurate diagnostic classification should be interpreted with caution (Figure 4).

### 3.4. Integrated Diagnostic Models for Sarcopenia: Multimodal Performance and Independent Predictors

The multimodal model combining SMI_T12CT, RF_CSA, and the 30 s squat test demonstrated good diagnostic accuracy for Sc, with an AUC of 0.826 (*p* < 0.001). The model achieved a sensitivity of 72.7% and a specificity of 84.8%. The positive predictive value (PPV) was 53.3%, while the negative predictive value (NPV) reached 92.9%, indicating a high capacity to correctly exclude Sc in non-affected individuals (Figure 5). The optimal cut-off points identified were 21.23 kg/m^2^ for SMI_T12CT, 2.38 cm^2^ for RF_CSA, and a threshold of 10 repetitions in the 30 s squat test. These findings support the integration of morphological, functional, and ultrasonographic techniques as a reliable approach for the comprehensive assessment of muscle health in clinical settings.

In the logistic regression model, lower skeletal muscle area at the T12 level (SMA_T12CT), older age, and reduced performance in the 30 s squat test were independently associated with increased odds of Sc. Specifically, each 1 cm^2^ increase in SMA_T12CT was associated with a 4% reduction in the odds of Sc (OR = 0.96, 95% CI: 0.92–0.99, *p* = 0.022). Age remained a strong risk factor, with each additional year increasing the odds by 23% (OR = 1.23, 95% CI: 1.07–1.47, *p* = 0.010). Functional impairment was also significant: each additional repetition in the 30 s squat test was associated with a 22% decrease in the likelihood of sarcopenia (OR = 0.78, 95% CI: 0.63–0.91, *p* = 0.007) (Table 7). The model demonstrated excellent discriminative performance (C-statistic = 0.875) and good calibration (Hosmer–Lemeshow *p* = 0.605), supporting its robustness and clinical applicability (Appendix A).

## 4. Discussion

This study provides new insights into the utility of thoracic CT-derived body composition parameters for diagnosing Sc and low muscle mass in patients with lung cancer. By leveraging routine imaging data at the T12 and T4 vertebral levels, we demonstrated that CT can serve as a reliable, opportunistic method for muscle evaluation when L3-level imaging is unavailable.

To our knowledge, this is one of the first studies to apply a comprehensive approach combining CT, BIVA, NU and functional testing for the assessment of Sc in lung cancer patients.

Cancer cachexia and Sc are frequently observed in lung cancer patients and are associated with increased postoperative complications, reduced tolerance to antineoplastic therapy, and decreased overall survival [7,13,25,26]. Early diagnosis of these conditions is crucial for optimizing clinical outcomes through nutritional interventions and tailored treatment approaches.

Consistent with the majority of studies, in our study, a higher percentage of men were affected by lung cancer. Our findings indicate that patients with Sc exhibit distinctive clinical and functional characteristics compared to those without Sc. Sarcopenic patients were significantly older, had lower muscle mass and function, a lower BMI and a higher ECOG [9,27]. Recent research has explored the feasibility of using thoracic levels, such as T12 or T4, for Sc assessment, demonstrating a strong correlation between thoracic levels and the established gold standard method at the L3 level, which is not included in routine studies performed on patients with LC [9,26,28,29,30]. Our findings reveal clear variations in the distribution and density of muscle and adipose tissue between patients, these characteristics had not been evaluated together previously. Sarcopenic individuals had a lower muscle mass and adippose tissue measured by BIVA, NU and CT scan. They also had poorer results on HGS and 30 s squat test, while no differences were found in TUG performance.

Our results indicate a moderate–high correlation between morphofunctional variables, in particular between parameters of BIVA (SMI, BCM), NU (RF_CSA and RF_Y_Axis), TC (SMI_T4CT and SMI_T12CT) and HGS, and the inverse of all these parameters with the Sc criterion EWGSOP2. We also found moderate to strong correlations between adiposity measures from BIVA (FMI), ultrasound (L-SAT, T-SAT), and CT (SAT, VAT, IMAT), supporting their coherence as fat-related indicators, although we have not found any studies in the literature that evaluate the correlation between measures of muscle and adipose tissue using different techniques. The integrated model combining SMI_T12CT, rectus femoris cross-sectional area (RF_CSA) by ultrasound, and the 30 s sit-to-stand test achieved high diagnostic accuracy (AUC = 0.826). This multimodal approach outperformed single-method assessments, reducing misclassification and improving overall sensitivity and specificity. The model’s discriminative ability (C-statistic = 0.875) supports the value of combining structural and functional parameters, aligning with the multidimensional nature of sarcopenia.

In our series, we observed that 53,8% of patients were classified as having low muscle mass when combining low ASMM and/or low ASMI. Regarding the diagnosis of Sc, it was observed that 20% patients presented criteria of low strength and low muscle mass according to the EWGSOP2 criteria. The prevalence of low skeletal muscle mass in patients with lung cancer established in the literature varies from 22.4 to 79.2%; these differences can be attributed to a multiple factors, including ethnic groups, data collection, dietary patterns, physical activity and socioeconomic status [7,31,32,33,34,35]. Correlations have been observed between RF-CSA Via NU and Pha Via BIVA in oncology patients [36,37,38], post-critical COVID-19 patients [39], idiopathic pulmonary fibrosis [14], and colorectal cancer [40].

T12-level measurements demonstrated superior performance compared to T4-level, particularly in women. A high correlation was also observed between SMI T12 and SMI T4, indicating coherence between muscle mass assessments at different vertebral levels, as indicates for Kaltenhouser [9]. Drestine et al. quantified reference values for Sc using lumbar and thoracic muscle CSA measures in a healthy US population, *N* = 604. SMA cutoffs for males/females, respectively, were 91.5/55.9 (T12), 141.7/91.2 (L3). SMI cutoffs for males/females, respectively, were 28.7/20.6 (T12) and 44.6/34.0 (L3) [26,41]. In our sample, Spanish lung cancer patients, predictive value to diagnose Sc in T12 with SMA for males/females was 89.39/54.37 at with SMI was 24.78/21.24; similar to that described by Drestine with a lower value in the men’s group. Meanwhile, the cut-off value of SMI_T4CT was 57.23/49.35 for Sc.

As explained in Table 6, our results suggest that while SMA and SMI at T4 level have a higher specificity, they lose a great deal of sensitivity at these cut-offs. However, SMI_T12CT provides a higher sensitivity, at the cost of losing some specificity, which may make it a more important parameter when diagnosing Sc. Wakefield et al. demonstrated that 24% of lung cancer patients (53/221) had Sc at T12 with a median lower SMI of 23.9 cm^2^/m^2^ [13]. In our Sc group, the average SMA at T12 was 62.09 and the average SMI 23.03, quite close to Wakefield references. Takamori et al. reported that the mean preoperative T12 SMI in lung cancer patients was 12.33 cm^2^/m^2^ in men and 11.22 cm^2^/m^2^ in women, and decreased, respectively, by 0.99 and 0.54 cm^2^/m^2^ after lung resection [42]. These findings highlight the critical need to establish gender-specific and population-specific consensus on T12 SMI cutoff values in the diagnosis and management of Sc among lung cancer patients. Cho et al. [43] studied 45 adult lung transplant recipients, and indicated that those with Sc, classified based on SMI_T12CT with a cut-off value of 28.07 cm^2^/m^2^, demonstrated worse postoperative survival.

The multivariable analysis highlights a distinct clinical profile associated with Sc in lung cancer patients, characterized by the convergence of structural muscle loss, advanced age, and reduced functional performance. These findings reinforce the multidimensional nature of Sc, which extends beyond muscle depletion to encompass broader physiological vulnerability. The strength of the predictive model supports its potential clinical utility for early identification of high-risk patients, thereby enabling more personalized nutritional interventions and treatment planning. Moreover, the inclusion of functional performance metrics—often underutilized in oncologic settings—underscores their prognostic value and relevance in assessing physiological reserve.

This study has several limitations. Firstly, the small sample size and limited number of sarcopenic patients may reduce the statistical power and limit the generalizability of the findings. Secondly, the study was conducted at a single center, which may restrict its external validity; therefore, its performance and applicability in broader clinical settings require further external validation in independent cohorts. In addition, the cohort was predominantly composed of male patients, reflecting the epidemiology of lung cancer, but this limits the applicability of the results to female populations. Another relevant limitation is the lack of standardized reference values for defining Sc using thoracic CT, and the modest AUC values and variability in sensitivity/specificity, raising concerns about T12CT as a standalone diagnostic tool. Finally, although the FocusedON^®^ platform offers promising capabilities for muscle segmentation, it remains an emerging methodology that requires further validation in clinical practice. There is also the possibility of residual confounding to the lack of information on variables that may affect the associations analyzed in our study.

The main strength of this study lies in its comprehensive approach to assessing body composition in lung cancer patients using multiple MFA techniques. This multifaceted method ensures a thorough evaluation of both muscle mass and function, providing robust data that highlight their interconnection. Notably, this integrative strategy enabled the development of a predictive model with relatively high discriminative ability, underscoring its potential utility in clinical risk stratification and personalized management.

Overall, our findings support the integration of morphofunctional tools—including imaging, ultrasound, and functional testing—for a more comprehensive and accurate assessment of muscle health in oncology patients. This approach aligns with the growing emphasis on personalized medicine and may contribute to better risk stratification, therapeutic planning, and outcome prediction in patients with lung cancer.

Future research should focus on larger multicenter cohorts with more diverse populations, establish international sex-specific reference standards for T12 indices, and validate proposed cut-offs externally, while also integrating artificial intelligence to enhance segmentation and promoting multimodal approaches that combine imaging, ultrasound, bioimpedance, and functional testing to improve diagnostic robustness.

## 5. Conclusions

In patients with lung cancer, CT-derived muscle indices at the T12 level, particularly SMI_T12CT, showed superior diagnostic accuracy over T4-derived measures for identifying low muscle mass and Sc. While T12CT shows promise as a practical alternative to L3CT, its moderate sensitivity and specificity highlight the importance of integrating it with morphofunctional tools such as BIVA, NU, and functional performance tests. These preliminary results support the integration of morphofunctional and imaging-based assessments to improve early detection of low muscle mass and Sc in routine clinical care.

## Figures and Tables

**Figure 1 cancers-17-03255-f001:**
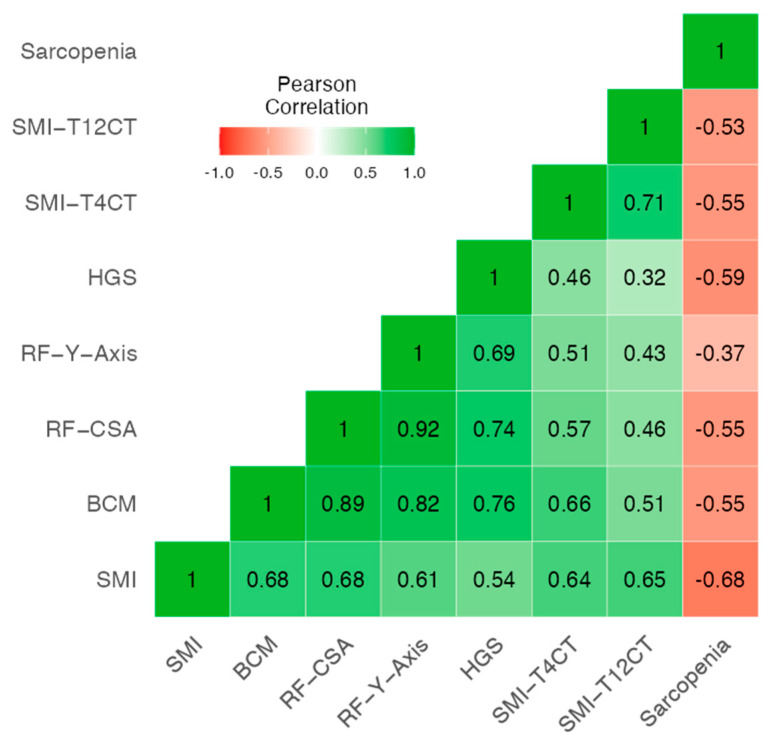
Heatmap of correlation between sarcopenia and morphofunctional variables including CT, HGS, NU and BIVA. Abbreviations: SMI: Skeletal muscle index by BIVA; BCM: Body cell mass; RF_CSA: Rectus femoris cross-sectional area (ultrasound); RF_Y_axis: rectus femoris *Y*-axis (vertical diameter); HGS: Handgrip strength; SMI_T4CT: Skeletal muscle index at the T4 level by computed tomography; SMI_T12CT: Skeletal muscle index at the T12 level by computed tomography.

**Figure 2 cancers-17-03255-f002:**
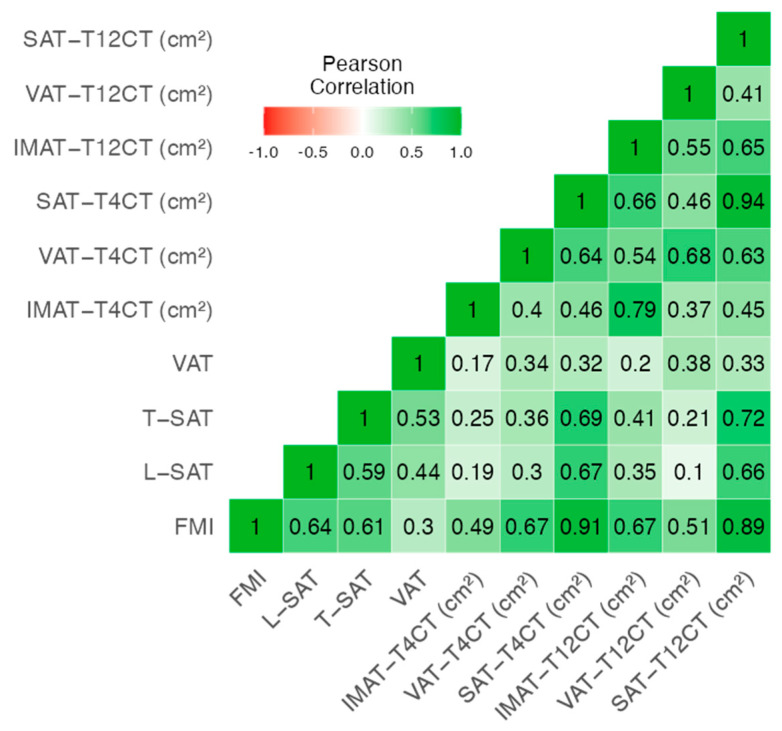
Heatmap of correlation between CT-derived adipose tissue parameters and morphofunctional fat-related variables. Abbreviations: FMI: Fat Mass Index; L-SAT: Lateral Subcutaneous Adipose Tissue (ultrasound); T-SAT: Total Subcutaneous Adipose Tissue (ultrasound); SAT: Subcutaneous Adipose Tissue (CT); VAT: Visceral Adipose Tissue (CT); IMAT: Intramuscular Adipose Tissue (CT).

**Figure 3 cancers-17-03255-f003:**
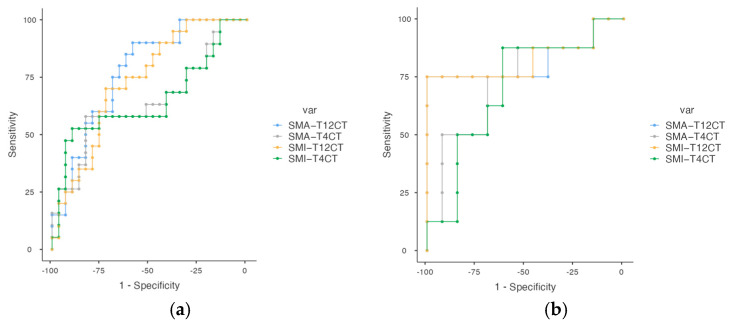
(**a**,**b**). ROC cut-off point for low muscle mass at T4 and T12 in lung cancer patients. Figure (**a**) corresponds to men and Figure (**b**) to women. Abbreviations: SMA_T12CT: skeletal muscle area at T12 level; SMI_T12CT: skeletal muscle index at T12 level; SMA_T4CT: skeletal muscle area at T4 level; SMI_T4CT: skeletal muscle index at T4 level.

**Figure 4 cancers-17-03255-f004:**
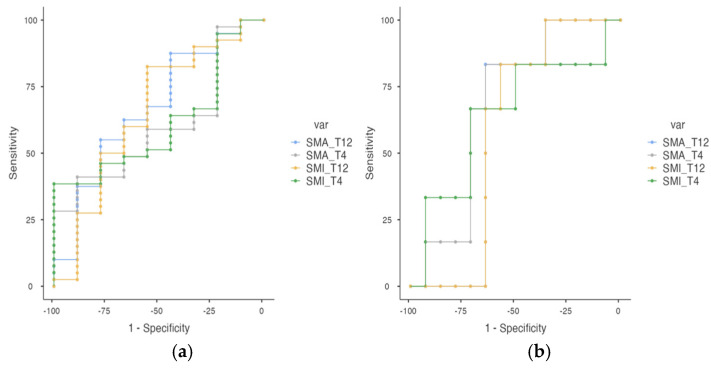
(**a**,**b**). ROC cut-off points for sarcopenia (EWGSOP2) at T4 and T12 levels in lung cancer patients. Figure (**a**) corresponds to men and Figure (**b**) to women. Abbreviations: SMA_T4CT: skeletal muscle area at T4 level by CT; SMA_T12CT: skeletal muscle area at T12 level by CT; SMI_T4CT: skeletal muscle index at T4 level by CT; SMI_T12CT: skeletal muscle index at T12 level by CT; AUC: area under the curve; PPV: positive predictive value; NPV: negative predictive value.

**Figure 5 cancers-17-03255-f005:**
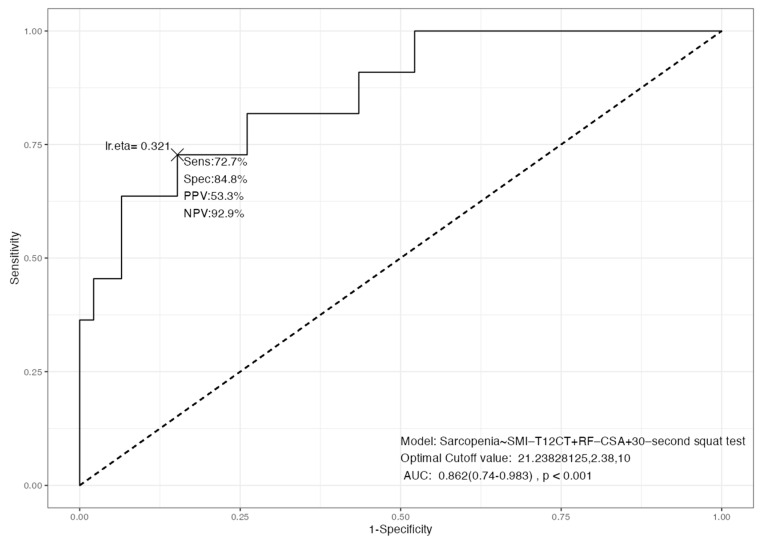
ROC curve for the multimodal model combining SMA_T12CT, RF_CSA, and 30 s squat test for sarcopenia detection. Abbreviations: SMI_T12CT: skeletal muscle index at T12 level by CT; RF_CSA: rectus femoris cross-sectional area (ultrasound); 30 s squat test: 30-second squat test; AUC: area under the curve; PPV: positive predictive value; NPV: negative predictive value.

**Table 1 cancers-17-03255-t001:** Comparison of Clinical and Treatment Characteristics Between Sarcopenic and Non-Sarcopenic Patients.

Variable	Category	Non-Sarcopenia (*N* = 64)	Sarcopenia (*N* = 16)	*p*-Value
Demographic				
Age	Mean ± SD	64.8 ± 9.58	71.8 ± 7.90	*p* = 0.009
Gender (Male)		71.9%	62.5%	*p* = 0.464
Clinicopathological				
TNM_T	T4	35.2%	11.1%	*p* = 0.48
TNM_N	N0	33.3%	5.6%	*p* = 0.68
TNM_M	M0	47.3%	14.5%	*p* = 0.88
Tumor Stage				
	Non-stage	14.0%	6.2%	*p* = 0.19
	Stage I	31.2%	43.7%	*p* = 0.54
	Stage II	6.2%	0.0%	*p* = 0.57
	Stage III	20.3%	25.0%	*p* = 0.73
	Stage IV	26.5%	31.2%	*p* = 0.97
Surgery	Yes	27.5%	5.0%	*p* = 0.15
Radiotherapy (RT)	Yes	26.6%	5.1%	*p* = 0.67
Chemotherapy (QT)	Yes	40.5%	5.1%	*p* = 0.12
Immunotherapy	Yes	22.8%	3.8%	*p* = 0.55
Tumor Classification	NSCLC	54.4%	12.7%	*p* = 0.35
	SCLC	2.5%	3.8%	
	Mesotelioma	3.8%	0.0%	
	NP	15.2%	3.8%	
	Neuroendocrino	2.5%	0.0%	
ECOG	0	42.3%	0.0%	*p* < 0.001
	1	31.0%	9.9%	
	2	8.5%	5.6%	
	3	0.0%	2.8%	

Data are summarized as means ± standard deviations (SDs), percentages, or absolute numbers. Groups were compared based on the diagnosis of sarcopenia, using Student’s *t*-test for normally distributed variables and the Mann–Whitney *U* test for non-normally distributed ones (normality assessed via the Shapiro–Wilk test). Categorical variables were compared using the chi-square test or Fisher’s exact test, depending on expected frequencies. Abbreviations: BMI: Body mass index; TNM: Tumor, node, metastasis; RT: Radiotherapy; QT: Chemotherapy; NSCLC: Non-small cell lung cancer; SCLC: Small-cell lung cancer; NP: Not specified (unclassified tumor type); ECOG: Eastern Cooperative Oncology Group performance status.

**Table 2 cancers-17-03255-t002:** Sarcopenia according to EWGSOP2 criteria.

*N* = 80			*p*-Value
Handgrip strength (kg)	Mean ± SD	30.1 ± 11.6	<0.001 *
Men	Mean ± SD	34.1 ± 10.2	
Women	Mean ± SD	20.8 ± 9.18	
ASMM (kg)		19.9 ± 4.92	<0.001 *
Men	Mean ± SD	21.9 ± 4.14	
Women	Mean ± SD	15.2 ± 2.9	
Low ASMM	*N* (%)	18 (22.5%)	<0.001 **
ASMI (kg/talla)	Mean ± SD	7.04 ± 1.34	<0.001 *
Men	Mean ± SD	7.46 ± 1.22	
Women	Mean ± SD	6.06 ± 1.09	
Low ASMI	*N* (%)	39 (48.8%)	0.182 **
Low muscle (Low ASMM/ASMI)	*N* (%)	43 (53.8%)	0.96 **
Dynapenia	*N* (%)	19 (23.8%)	<0.87 **
Sarcopenia (dynapenia + low muscle)	*N* (%)	16 (20%)	0.47 **

* *p*-values refer to statistical comparisons by sex. ** *p*-values refer to statistical comparisons high or low percentages of the different criteria for determining sarcopenia. Abbreviations: ASMM: Appendicular skeletal muscle mass; ASMI: Appendicular skeletal muscle mass index.

**Table 3 cancers-17-03255-t003:** Morphofunctional and Body Composition Parameters.

Variable	Non-Sarcopenia (*N* = 64)	Sarcopenia (*N* = 16)	*p*-Value
BIVA	Mean ± SD	Mean ± SD	
BMI (kg/m^2^)	27.4 ± 4.93	23.2 ± 2.96	0.002
PA (º)	4.72 ± 0.86	4.06 ± 0.93	0.009
SPA	−1.39 ± 0.90	−1.86 ± 0.89	0.086
Rz	516.5 ± 92.3	573.8 ± 70.8	0.023
Xc	42.5 ± 9.92	40.8 ± 10.4	0.533
BCM (kg)	25.2 ± 6.65	19.4 ± 5.83	0.002
ASMM (kg)	20.7 ± 4.89	16.7 ± 3.69	0.003
FFM (kg)	54.2 ± 10.19	46.4 ± 8.00	0.005
TBW (kg)	41.2 ± 8.48	35.1 ± 5.65	0.009
ECW (kg)	21.7 ± 4.69	20.1 ± 3.36	0.238
FM (kg)	23.1 ± 8.73	16.7 ± 5.94	0.007
NAK	1.24 ± 0.26	1.47 ± 0.40	0.006
Hydration (%)	75.7 ± 3.99	76.1 ± 5.11	0.814
Nutrition	712.1 ± 213.0	584.4 ± 161.6	0.028
SMI (kg)	9.16 ± 1.87	7.94 ± 1.25	0.016
Echography exploration			
RF_CSA	3.83 ± 1.30	2.65 ± 0.73	0.002
RF_Cir	9.01 ± 1.34	8.14 ± 1.04	0.030
RF_X_Axis	3.59 ± 0.59	3.37 ± 0.50	0.224
RF_Y_Axis	1.19 ± 0.35	0.84 ± 0.18	<0.001
L_SAT (cm)	0.83 ± 0.53	0.53 ± 0.28	0.048
RF_Cont (cm)	1.47 ± 1.43	1.08 ± 0.29	<0.001
T-SAT (cm)	1.53 ± 0.76	1.05 ± 0.46	0.027
S-SAT (cm)	0.68 ± 0.39	0.51 ± 0.22	0.176
VAT (cm)	0.53 ± 0.32	0.34 ± 0.21	0.036
Functional test			
HGS (kg)	33.6 ± 9.79	16.5 ± 7.69	<0.001
TUG (second)	6.91 ± 2.66	6.28 ± 5.07	0.544
30 s squat test	11.09 ± 6.83	6.5 ± 6.00	0.036

Data are summarized as means ± standard deviations, percentages, or absolute numbers. Groups were compared based on the diagnosis of sarcopenia using Student’s *t*-test for normally distributed variables and the Mann–Whitney *U* test for non-normally distributed ones (normality assessed via the Shapiro–Wilk test). Categorical variables were compared using the chi-square test or Fisher’s exact test, depending on expected frequencies. Abbreviations: BMI: Body Mass Index; PA: Phase Angle; SPA: Standardized Phase Angle; Rz: Resistance; Xc: Reactance; BCM: Body Cell Mass; ASMM: Appendicular Skeletal Muscle Mass; FFM: Fat-Free Mass; TBW: Total Body Water; ECW: Extracellular Water; FM: Fat Mass; NAK: Sodium–Potassium Ratio; Hydration: Hydration Percentage; Nutrition: Nutrition Index; SMI: Skeletal Muscle Index (BIVA); RF_CSA: Rectus Femoris Cross-Sectional Area; RF_Y_Axis: Rectus Femoris Y axis; L_SAT: Leg Subcutaneous Adipose Tissue; RF_Cont: Rectus Femoris Muscle Contraction; T-SAT: Total Abdominal Subcutaneous Adipose Tissue; S-SAT: Superficial Abdominal Subcutaneous Adipose Tissue; VAT: Visceral Adipose Tissue; TUG: Timed Up and Go test.

**Table 4 cancers-17-03255-t004:** Differences in body composition parameters by T4-CT and T12-CT according to sarcopenia.

Variable	Non-Sarcopenia (*N* = 52) Mean ± SD	Sarcopenia (*N*= 15) Mean ± SD	*p*-Value
Muscle_T4CT (%)	28.00 ± 6.89	29.64 ± 7.90	0.433
SMA_T4CT (cm^2^)	147.7 ± 39.3	123.7 ± 31.1	0.033
SMI_T4CT (cm^2^/m^2^)	51.8 ± 11.3	46.6 ± 10.6	0.120
Muscle_T4CT (UH)	43.95 ± 10.43	40.04 ± 12.77	0.227
IMAT_T4CT (%)	2.05 ± 0.94	2.35 ± 1.02	0.318
IMAT_T4CT (cm^2^)	11.08 ± 5.78	10.69 ± 5.39	0.923
IMAT_T4CT (UH)	−68.78 ± 8.05	−68.00 ± 4.97	0.762
VAT_T4CT (%)	6.33 ± 3.36	7.41 ± 3.33	0.273
VAT_T4CT (cm^2^)	35.93 ± 22.93	35.37 ± 19.91	0.932
VAT_T4CT (UH)	−94.79 ± 7.59	−94.10 ± 5.54	0.743
SAT_T4CT (%)	28.13 ± 9.54	24.75 ± 9.05	0.224
SAT_T4CT (cm^2^)	153.90 ± 73.34	115.26 ± 52.38	0.092
SAT_T4CT (UH)	−96.29 ± 13.50	−92.13 ± 11.35	0.084
Muscle_T12CT (%)	11.63 ± 2.76	12.20 ± 4.20	0.533
SMA_T12CT (cm^2^)	82.64 ± 26.29	62.09 ± 12.65	0.008
SMI_T12CT (cm^2^/m^2^)	29.12 ± 8.48	23.03 ± 4.21	0.015
Muscle_T12CT (UH)	36.53 ± 16.00	32.74 ± 14.94	0.414
IMAT_ T12CT (%)	1.25 ± 0.70	1.26 ± 0.60	0.822
IMAT_T12CT (cm^2^)	9.38 ± 6.25	7.86 ± 5.44	0.356
IMAT_T12CT (UH)	−64.02 ± 11.57	−64.41 ± 6.26	0.517
VAT_T12CT (%)	15.17 ± 9.24	16.10 ± 8.96	0.730
VAT_T12CT (cm^2^)	120.79 ± 84.78	107.29 ± 80.41	0.583
VAT_T12CT (UH)	−93.02 ± 8.42	−92.45 ± 6.68	0.811
SAT_T12CT (%)	14.70 ± 7.04	11.96 ± 4.09	0.198
SAT_T12CT (cm^2^)	108.44 ± 68.13	71.59 ± 32.32	0.047
SAT_T12CT (UH)	−93.55 ± 16.19	−87.17 ± 14.36	0.040

Data are summarized as means ± standard deviations. Groups were compared based on the diagnosis of sarcopenia using Student’s *t*-test or Mann–Whitney *U* test, depending on data normality (assessed with the Shapiro–Wilk test). Abbreviations: T4CT and T12CT: Thoracic vertebral levels 4 and 12 on CT imaging; MUSCLE: Skeletal muscle area expressed as percentage of the total cross-sectional tissue; SMA: Skeletal Muscle Area (cm^2^); SMI: Skeletal Muscle Index (cm^2^/m^2^); HU: Hounsfield Units, indicating tissue density; IMAT: Intramuscular Adipose Tissue; VAT: Visceral Adipose Tissue; SAT: Subcutaneous Adipose Tissue.

**Table 5 cancers-17-03255-t005:** Predictive value to diagnose low muscle mass in T4CT and T12CT.

Variable	Cut-Off	AUC	Sensitivity (%)	Specificity (%)	PPV (%)	NPV (%)
SMA_T4CT (Men)	169.89	0.644	57.89	82.76	68.75	75.00
SMA_T4CT (Women)	130.02	0.736	57.14	92.31	80.00	80.00
SMA_T12CT (Men)	80.34	0.772	90.00	58.62	60.00	89.47
SMA_T12CT (Women)	70.47	0.791	71.43	100.00	100.00	86.67
SMI_T4CT (Men)	59.05	0.650	52.63	89.66	76.92	74.29
SMI_T4CT (Women)	41.69	0.714	85.71	61.54	54.55	88.89
SMI_T12CT (Men)	31.98	0.733	70.00	72.41	63.64	77.78
SMI_T12CT (Women)	28.23	0.802	71.43	100.00	100.00	86.67

Abbreviations: SMA_T12CT: Skeletal muscle area at T12 level; SMI_T12CT: Skeletal muscle index at T12 level; SMA_T4CT: Skeletal muscle area at T4 level; SMI_T4CT: Skeletal muscle index at T4 level.

**Table 6 cancers-17-03255-t006:** Predictive value to diagnose sarcopenia (EWGSOP2) at T4CT and T12CT.

Variable	Cut-Off	AUC	Sensitivity (%)	Specificity (%)	PPV (%)	NPV (%)
SMA_T4CT (Men)	165.76	0.598	41.03	88.89	94.12	25.81
SMA_T4CT (Women)	105.27	0.631	83.33	64.29	50.00	90.00
SMA_T12CT (Men)	89.39	0.669	55.00	77.78	91.67	28.00
SMA_T12CT (Women)	54.37	0.595	83.33	64.29	50.00	90.00
SMI_T4CT (Men)	57.23	0.610	38.46	100.00	100.00	27.27
SMI_T4CT (Women)	49.35	0.643	66.67	71.43	50.00	83.33
SMI_T12CT (Men)	24.78	0.653	82.50	55.56	89.19	41.67
SMI_T12CT (Women)	21.24	0.583	83.33	57.14	45.45	88.89

Abbreviations: SMA_T4CT: skeletal muscle area at T4 level by CT; SMA_T12CT: skeletal muscle area at T12 level by CT; SMI_T4CT: skeletal muscle index at T4 level by CT; SMI_T12CT: skeletal muscle index at T12 level by CT; AUC: area under the curve; PPV: positive predictive value; NPV: negative predictive value.

**Table 7 cancers-17-03255-t007:** Multivariable Logistic Regression Identifying Independent Predictors of sarcopenia.

Dependent: Sarcopenia		No	Yes	OR (Univariable)	OR (Multivariable)
SMA_T12CT	Mean ± SD	83.8 ± 26.9	69.0 ± 23.6	0.98 (0.95–1.00, *p* = 0.106)	0.96 (0.92–0.99, *p* = 0.022)
Age	Mean ± SD	65.8 ± 9.5	72.5 ± 8.0	1.10 (1.01–1.23, *p* = 0.046)	1.23 (1.07–1.47, *p* = 0.010)
30 s squat test	Mean ± SD	11.2 ± 6.9	6.0 ± 6.0	0.89 (0.80–0.98, *p* = 0.031)	0.78 (0.63–0.91, *p* = 0.007)

Abbreviations: SMA_T12CT skeletal muscle area at the T12 vertebral level assessed by computed tomography; OR odds ratio; CI indicates confidence interval; SD standard deviation.

## Data Availability

The original contributions presented in the study are included in the article, further inquiries can be directed to the corresponding authors.

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
