# Peer review of "AI-Assistance Body Composition CT at T12 and T4 in Lung Cancer: Diagnosing Sarcopenia, and Its Correlation with Morphofunctional Assessment Techniques"

_cancers, 2025, doi:10.3390/cancers17193255_

Round 1

Reviewer 1 Report

Comments and Suggestions for Authors

This is an interesting and relevant paper. I have several concerns regarding the methods and reporting that the authors should address.

Sample size and power: Although the authors acknowledge the small sample size in the discussion, given that this is a prospective study, it would be helpful to know whether a formal sample size calculation was performed to ensure adequate power for the study’s objectives.

Introduction and discussion: Both sections are quite long and, at times, not well supported by references. The authors should streamline these sections and provide appropriate citations where needed.

Repetition: Lines 175–185 appear to be repeated and should be revised for clarity.

Study Population: In the methods, the authors describe the study population as “lung cancer,” but Table 1 specifies subtypes (NSCLC, SCLC, and others). Please clarify this in the methods section by indicating how lung cancer subtypes were identified and included. This will help readers better understand the study population and ensure consistency across sections.

Multivariable analysis: The authors should clearly specify which variables were adjusted for in the multivariable logistic regression models.

Table 1: Please add cancer stage information to Table 1, as this is an important clinical characteristic. I am also wondering whether cancer stage may be related to sarcopenia risk, perhaps through associations with cachexia or disease burden. Even if not directly tied to sarcopenia, the stage could still provide useful context for interpreting the study population.

Interpretation of findings: The results suggest that T12CT may be an alternative to abdominal CT for sarcopenia screening, particularly when abdominal imaging is unavailable. However, the small sample size, low AUC values, and inconsistent sensitivity/specificity raise concerns that T12CT alone may not provide sufficient accuracy for diagnostic purposes. The discussion should better reflect these limitations.

Minor: There is a typo in the title — “Assistence” should be corrected to “Assistance.

Author Response

Response to Reviewer 1 Comments

 We sincerely thank the reviewer for their thoughtful and constructive feedback. Below, we address each point raised, aiming to clarify and improve our manuscript accordingly.

Comments 1: Sample size and power: Although the authors acknowledge the small sample size in the discussion, given that this is a prospective study, it would be helpful to know whether a formal sample size calculation was performed to ensure adequate power for the study’s objectives.

Response 1: We appreciate this important observation. A formal sample size estimation was conducted prior to study initiation to ensure the feasibility and relevance of our data. Based on an expected moderate effect size (Cohen’s d = 0.6), an alpha risk of 0.05, and a beta risk of 0.2 (power = 80%), we estimated that approximately 72 patients would be needed to detect meaningful differences between sarcopenic and non-sarcopenic groups. Although this calculation was not initially included in the manuscript, we enrolled 80 patients to ensure adequate statistical power and account for potential losses during follow-up.

Comments 2:  Introduction and discussion: Both sections are quite long and, at times, not well supported by references. The authors should streamline these sections and provide appropriate citations where needed.

Response 2: Thank you for highlighting this. We have revised both the Introduction and Discussion sections, condensing the content to enhance clarity and focus. Additionally, we have added missing citations where appropriate to better support the narrative and ensure academic rigor.

Comments 3:  Repetition: Lines 175–185 appear to be repeated and should be revised for clarity.

Response 3:  We acknowledge this oversight. The repeated section has now been revised for clarity and the redundancy eliminated.

Comments 4:  Study Population: In the methods, the authors describe the study population as “lung cancer,” but Table 1 specifies subtypes (NSCLC, SCLC, and others). Please clarify this in the methods section by indicating how lung cancer subtypes were identified and included. This will help readers better understand the study population and ensure consistency across sections.

 Response 4:  We thank the reviewer for this helpful observation. As suggested, we have now clarified in the Methods section that lung cancer subtypes were determined through histological confirmation at diagnosis and classified as non-small cell lung cancer (NSCLC), small cell lung cancer (SCLC), or other subtypes, based on standard pathology and oncological criteria. This information has been added to ensure consistency with Table 1 and to improve the clarity and understanding of the study population.

Comments 5:  Multivariable analysis: The authors should clearly specify which variables were adjusted for in the multivariable logistic regression models.

Response 5:  We have clarified in the revised manuscript that the multivariable logistic regression model was designed as an exploratory approach to identify predictors of sarcopenia based on novel assessment techniques. Specifically, the model included three variables: skeletal muscle area at the T12 vertebral level (SMA_T12CT), age, and the number of repetitions in the 30-second squat test. These were chosen as alternative markers of low muscle mass and poor physical performance, respectively, distinct from the EWGSOP2 operational criteria.

Our aim was to evaluate the diagnostic potential of methods that are feasible in routine clinical practice, such as opportunistic CT imaging and a simple functional test. The final model showed that lower SMA_T12CT (OR = 0.96, 95% CI: 0.92–0.99, p = 0.022), older age (OR = 1.23, 95% CI: 1.07–1.47, p = 0.010), and reduced performance in the 30-second squat test (OR = 0.78, 95% CI: 0.63–0.91, p = 0.007) were independently associated with increased odds of sarcopenia. The model demonstrated excellent discriminative performance (C-statistic = 0.875) and good calibration (Hosmer–Lemeshow p = 0.605), suggesting it may be a clinically useful tool in this population.

Comments 6:  Table 1: Please add cancer stage information to Table 1, as this is an important clinical characteristic. I am also wondering whether cancer stage may be related to sarcopenia risk, perhaps through associations with cachexia or disease burden. Even if not directly tied to sarcopenia, the stage could still provide useful context for interpreting the study population.

Response 6:  We agree that tumor stage is a relevant clinical characteristic that may provide context regarding disease severity and potential associations with sarcopenia. As requested, we have now included tumor stage information in Table 1. Although no statistically significant differences were observed, sarcopenic patients showed a trend toward more advanced stages (Stage III–IV), which may suggest a link with disease burden or cachexia.

Comments 7:  Interpretation of findings: The results suggest that T12CT may be an alternative to abdominal CT for sarcopenia screening, particularly when abdominal imaging is unavailable. However, the small sample size, low AUC values, and inconsistent sensitivity/specificity raise concerns that T12CT alone may not provide sufficient accuracy for diagnostic purposes. The discussion should better reflect these limitations.

Response 7:  We have now revised the discussion to more clearly acknowledge the limitations related to the diagnostic performance of T12CT. Specifically, we emphasize that although T12CT may be a practical alternative when abdominal imaging is unavailable, its relatively low AUC and the trade-off between sensitivity and specificity—particularly at the T4 level—limit its standalone utility for definitive sarcopenia diagnosis. These points have been integrated into the discussion to provide a more balanced interpretation of our findings and highlight the need for further validation in larger cohorts.

Comments 8:  Minor: There is a typo in the title — “Assistence” should be corrected to “Assistance.

Response 8:  Thank you for pointing this out. The typo in the title has been corrected from “Assistence” to “Assistance.”

Reviewer 2 Report

Comments and Suggestions for Authors

Review Comments

This article aims to assess the usefulness of T12-CT and T4-CT, obtained opportunistically from routine imaging, in identifying Sc and low muscle mass, and their correlation with morphofunctional tools such as BIVA, NU, TUG, and the 30-second chair stand test, which may support the integration of these tools for a more comprehensive and accurate assessment of muscle health in oncology patients. Several suggestions for improving this manuscript are listed below:

  1. The summarization of this study in the last paragraph of the Introduction section (Line 136-138) should be further enriched.
  2. Literature indexing should be avoided in the section heading (e.g., Line 186, Line 212, etc.).
  3. The alphabet number (“a” and “b”) in Fig. 3 should be moved to the upper left corner of each subgraph.
  4. It seems that Fig. 4b has been compressed because it looks narrower than Fig. 4a.
  5. The words in Fig. 5 should be further enlarged to increase readability.
  6. The author mentioned “Fig. 2S” in Line 465, while no relevant images can be found throughout the entire text.
  7. Various limitations were mentioned in Line 572-585, and how does the author plan to avoid them?
  8. Why the heading of Section 6 is “Patent”? No relevant content can be found in this section.
  9. The format of the references should be further improved.
  10. Several minor errors: 1) Two blank spaces should be added at the beginning of the second paragraph in Introduction section (Line 80-88); 2) the full stop should be removed from the section heading of Section 2.2.6 (Line 237); 3) the square symbol should be superscript in Line 556-557.

Author Response

Response to Reviewer 2 Comments

Comments 1: The summarization of this study in the last paragraph of the Introduction section (Line 136-138) should be further enriched.

Response 1: Thank you for the suggestion. As recommended, we have enriched the final paragraph of the Introduction (Lines 168–181) to provide a more comprehensive overview of the study’s rationale, objectives, and methodological approach.

Comments 2: Literature indexing should be avoided in the section heading (e.g., Line 186, Line 212, etc.).

Response 2: As suggested, we have removed the literature indexing from the section headings , ensuring that they now follow the journal’s formatting standards.

Comments 3: The alphabet number (“a” and “b”) in Fig. 3 should be moved to the upper left corner of each subgraph.

Response 3: The labels “a” and “b” in Figure 3 have now been repositioned to the upper left corner of each subgraph to improve clarity and presentation.

Comments 4: It seems that Fig. 4b has been compressed because it looks narrower than Fig. 4a.

Response 4: Figure 4b has been resized to match the proportions of Figure 4a to ensure visual consistency between the subfigures.

Comments 5: The words in Fig. 5 should be further enlarged to increase readability.

Response 5: The text in Figure 5 has been enlarged to improve readability and ensure clearer visualization of the data presented.

Comments 6: The author mentioned “Fig. 2S” in Line 465, while no relevant images can be found throughout the entire text.

Response 6: The figure referred to as “Fig. 2S” in Line 465 corresponds to a supplementary figure provided in the Supplementary Material of the manuscript. It presents a forest plot displaying the odds ratios (ORs), 95% confidence intervals, and p-values from the multivariable logistic regression analysis identifying predictors of sarcopenia.

Comments 7: Various limitations were mentioned in Line 572-585, and how does the author plan to avoid them?

Response 7: We appreciate the reviewer’s thoughtful comment. As outlined in the Discussion section, the limitations of our study primarily relate to the relatively small sample size and the single-center design. To address these issues, we have already initiated a larger, multicenter study aimed at validating our findings in a broader and more diverse population. This ongoing effort will help to confirm the robustness of our proposed morphofunctional screening model and improve its generalizability to clinical practice.

Comments 8: Why the heading of Section 6 is “Patent”? No relevant content can be found in this section.

Response 8: The inclusion of “Patent” as Section 6 was an oversight resulting from the use of the Cancers journal template, where this heading was not removed. We confirm that there are no patents associated with this work, and the section has now been deleted from the manuscript.

Comments 9:The format of the references should be further improved.

Response 9: The reference list has been carefully revised and reformatted to ensure full compliance with the journal’s citation style and formatting guidelines.

Comments 10: Several minor errors: 1) Two blank spaces should be added at the beginning of the second paragraph in Introduction section (Line 80-88); 2) the full stop should be removed from the section heading of Section 2.2.6 (Line 237); 3) the square symbol should be superscript in Line 556-557.

Response 10: All the indicated formatting issues have been carefully corrected: (1) two blank spaces were added at the beginning of the second paragraph in the Introduction section, (2) the full stop was removed from the heading of Section 2.2.6, and (3) the square symbol was changed to superscript format in Line 556–557.

Round 2

Reviewer 1 Report

Comments and Suggestions for Authors

Could the authors briefly include the sample size calculation in the methods section? 

Author Response

We thank the reviewer for this pertinent suggestion. As recommended, we have now included a brief description of the sample size calculation in the Materials and Methods section (2.1. Study Setting). Specifically, we state that a formal sample size estimation was performed prior to study initiation, based on a moderate effect size (Cohen’s d = 0.6), an alpha error of 0.05, and a beta error of 0.2 (power = 80%). This yielded a required minimum of 72 participants to detect meaningful differences between sarcopenic and non-sarcopenic groups. To ensure adequate power and account for potential losses, we enrolled a total of 80 patients.
